# A light-driven artificial flytrap

Owies M. Wani[1], Hao Zeng[1] & Arri Priimagi[1]

The sophistication, complexity and intelligence of biological systems is a continuous source of inspiration for mankind. Mimicking the natural intelligence to devise tiny systems that are capable of self-regulated, autonomous action to, for example, distinguish different targets, remains among the grand challenges in biomimetic micro-robotics. Herein, we demonstrate an autonomous soft device, a light-driven flytrap, that uses optical feedback to trigger photomechanical actuation. The design is based on light-responsive liquid-crystal elastomer, fabricated onto the tip of an optical fibre, which acts as a power source and serves as a contactless probe that senses the environment. Mimicking natural flytraps, this artificial flytrap is capable of autonomous closure and object recognition. It enables self-regulated actuation within the fibre-sized architecture, thus opening up avenues towards soft, autonomous small-scale devices.

[1] Laboratory of Chemistry and Bioengineering, Tampere University of Technology, PO Box 541, FI-33101 Tampere, Finland. Correspondence and requests for materials should be addressed to H.Z. (email: hao.zeng@tut.fi) or to A.P. (email: arri.priimagi@tut.fi).

Ranging from multiscale and functional self-assemblies to morphology control, complex locomotion strategies and autonomous feedback dynamics, nature has provided us with numerous fascinating examples for biomimetic research[1–5]. Recently, great efforts have been made in mimicking organic species to devise micro-robotic systems with novel functionalities and accessibility to increasingly challenging spaces[6–10]. However, mimicking the intelligence of natural species in artificial systems, that is, realization of devices that act autonomously and are capable of adapting to unexpected environmental changes, is a long-standing challenge. To date, the mainstream research on artificial intelligence is based on programming, hence relying on computer-controlled electronic actuation[11]. However, incorporation of complex computing circuitry, power sources and electrically driven actuators into miniaturized robotic systems is challenging, and other approaches are needed to devise smart robotic actuators.

Soft-matter-based micro-robotics is a nascent field in biomimetic research, with significant progress in the past few years[12,13]. Unlike conventional hard machines with rigid arms, soft robots provide natural safety and human-friendly contact. The flexible and miniaturized body also offers additional freedom for complex motion and adaptation to environmental confinement[14,15]. Several powering strategies have been developed for soft robotics, such as pneumatic networks[12], light illumination[13,15] and chemical reactions[16]. Automation is a particular challenge in soft devices, since the ways of powering and control need to be completely re-thought. A limited number of attempts have been made to create self-oscillating devices[16–19] – autonomous systems that could extract energy from a constant source and transfer it to cyclic mechanical work. Most of them employ stimuli-responsive materials such as light-responsive polymers[17,18] or chemically responsive hydrogels[19], in which a non-equilibrium oscillation occurs once a positive feedback mechanism[20] is established between deformed geometry and external stimulus field.

Here, we demonstrate an intelligent gripping device, a light-driven artificial flytrap. The name stems from the fact that the device is capable of mimicking the motions of natural flytrap (*Dionaea muscipula*)[21,22], by performing autonomous closure action (gripping) and self-recognition between different micro-objects by sensing their physical properties.

## Results

**System concept**. Venus Flytrap[21] (Fig. 1a,b) has fascinated scientists due to its unique features such as automatic closure of leaf upon mechanical stimulation, sub-second-scale fast actuation, and ability to distinguish insects and other prey from random particles like dust. Inspired by the flytrap, we propose a strategy to realize an artificial, fully light-fueled micro-device. We use a thin layer of light-responsive liquid-crystal elastomer (LCE) as an actuating material. LCEs are smart materials that can undergo huge shape changes in response to various external stimuli such as light, heat, or electric field[23–26]. Among the different stimuli, light is in many ways particularly attractive. It provides a clean, contactless energy source, and its properties (wavelength, intensity, polarization) can be optimized for a specific target with high spatial and temporal resolution. Therefore, several light-powered, LCE-based robotic[27–30] as well as tunable photonic[31–33] systems have been proposed. In essence, the light-induced actuation is based on controlling the molecular alignment within the liquid-crystal polymer network[34]. By choosing a splayed molecular alignment across the actuator thickness, light-induced alignment changes give rise to different strains within one monolithic layer: an expansion on one surface and contraction on the other. As a consequence, pronounced light-induced bending deformation occurs[35] (insets in Fig. 1c,d).

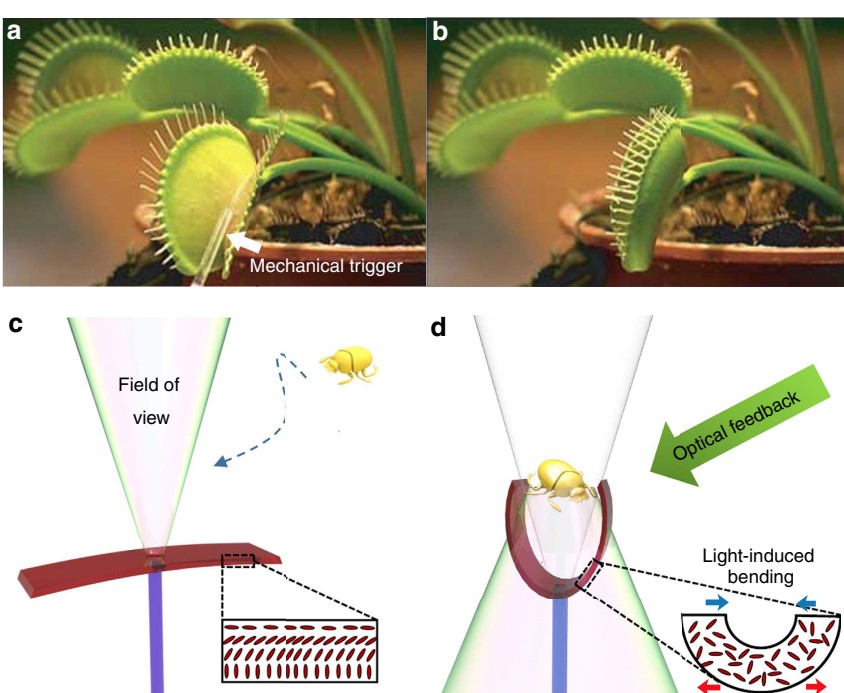

**Figure 1 | Flytrap-inspired light-powered soft robot.** (**a**) A Venus flytrap at its open stage, (**b**) closes upon mechanical stimulation. Reprinted with permission from ref. 21. (**c**) Schematic drawing of the light-triggered artificial flytrap at its open stage, when no object has entered its field of view. No light is back-reflected to the LCE actuator, which remains in the open stage. (**d**) The flytrap closes when an object enters its field of view and causes optical feedback to the LCE actuator. Light-induced bending of the LCE leads to closure action, thus capturing the object. The insets of **c** and **d** show the schematic molecular orientation in LCE actuator at the open and closed stages.

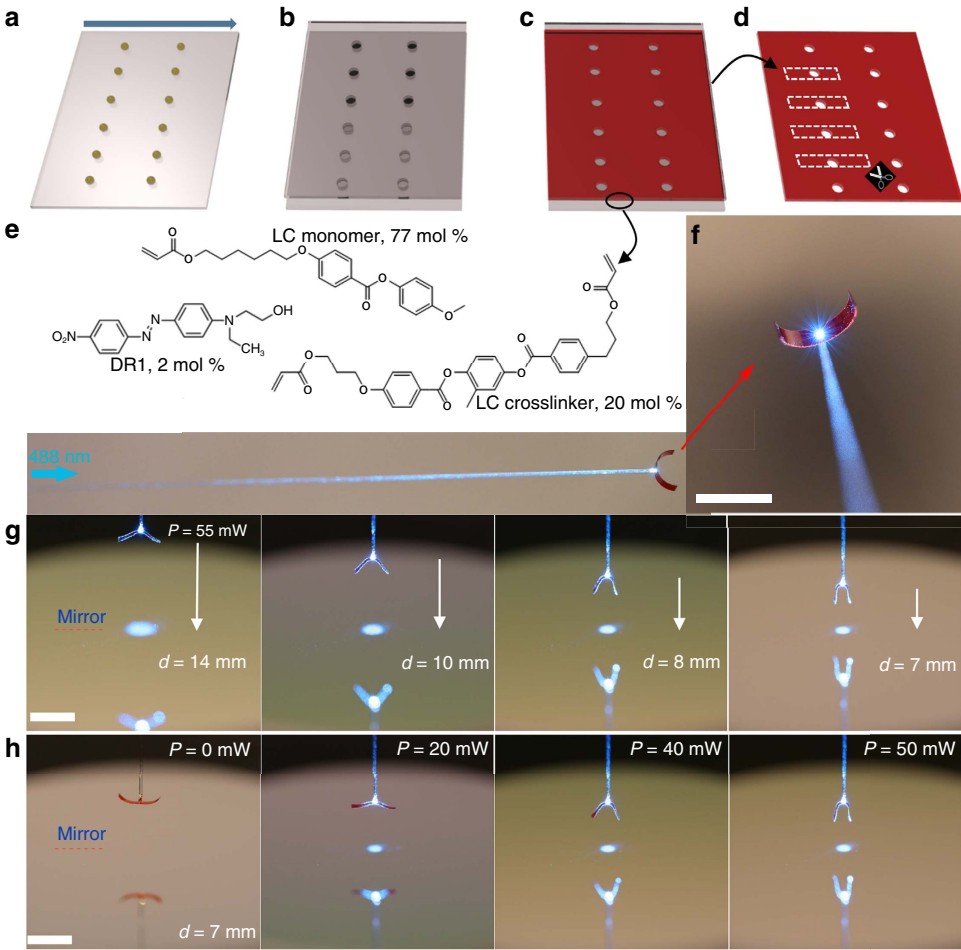

**Figure 2 | Realization of the autonomous gripper.** Schematic pictures of the fabrication process: (**a**) Arrays of UV curable resin are put on a glass substrate coated with rubbed PVA (the arrow indicates the rubbing direction). (**b**) A 20 μm LC cell is prepared by placing another glass slide coated with homeotropic alignment layer on the top, and subsequent curing with UV light to solidify the resin. (**c**) Liquid crystal monomers are infiltrated into the cell, and then UV-polymerized at 30 °C. (**d**) The cell is opened, and strips of LCE actuator are cut out from the substrate along the rubbing direction. (**e**) Chemical composition of the LC monomer mixture. (**f**) Optical images of the fabricated gripper after connecting to the fibre tip. (**g**) Gripper automatically closes while approaching to the mirror surface with a constant power of 55 mW. (**h**) At a constant distance $d = 7$ mm, gripper can be switched between closed and open stages by manually tuning the light power (0, 20, 40, 50 mW). All scale bars correspond to 5 mm.

To make this bending deformation to sense the environment, we integrated the LCE actuator with an optical fibre. More specifically, we fabricated the splay-aligned LCE actuator onto the tip of the fibre, leaving a transparent window in the center, through which light is emitted. The emission cone determines the field of view of the device, which continuously probes the space in front (Fig. 1c). When an object enters into the field of view and produces enough optical feedback (reflected/scattered light), the LCE bends towards the object (closure action), eventually capturing it (Fig. 1d). The optical feedback is determined by the reflectance/scattering intensity of the object, and therefore, the artificial flytrap may exhibit distinct actuation behavior when meeting different targets. Compared with other flytrap-like devices reported to date[36,37], this is the first miniature device that mimics the intelligent features of the Venus flytrap, while the mechanical motion is triggered by light-induced bending of the LCE actuator, not elastic instability[21] as in the case of the Venus flytrap.

**System realization.** A glass slide coated with polyvinyl alcohol (PVA) was firstly rubbed unidirectionally. Arrays of droplets of UV-curable resin (few nl each) were put onto the surface, separated by ∼1 cm distance (Fig. 2a). Another glass slide coated

with homeotropic alignment layer was placed on top of the one containing the droplets, maintaining 20 μm separation between the two slides by using spacer beads. Subsequently, a UV lamp was used to cure the resin and form an LC cell (Fig. 2b). Liquid-crystal monomer mixture was then infiltrated at 70 °C and polymerized after cooling to the nematic phase at 30 °C. The mixture contained commercially available acrylate-functionalized LC monomers and cross-linkers (Fig. 2e). As a light-responsive unit, we used Dispersed Red 1, added as a dopant into the LCE network, similar to previous reports[38,39]. This system allows for more than 80% light absorption at the excitation wavelength (488 nm) within the 20 μm thick splay-aligned layer. A UV lamp was used to cure the mixture, forming a well-aligned LCE film with an array of transparent windows (Fig. 2c). The order parameter of the LCE film, as deduced from polarized absorption measurements, was *ca.* 0.6 (see Supplementary Note 1 and Supplementary Fig. 5 for further details). The cell was opened and strips with transparent windows at their central positions were cut along the rubbing direction (Fig. 2d). Finally, the optical flytrap was formed by fixing the tip of a multi-mode optical fibre to the center of the LCE strip by using another droplet of cured resin (attached to the homeotropically-aligned surface). Light from a 488 nm laser was coupled to the other end of the

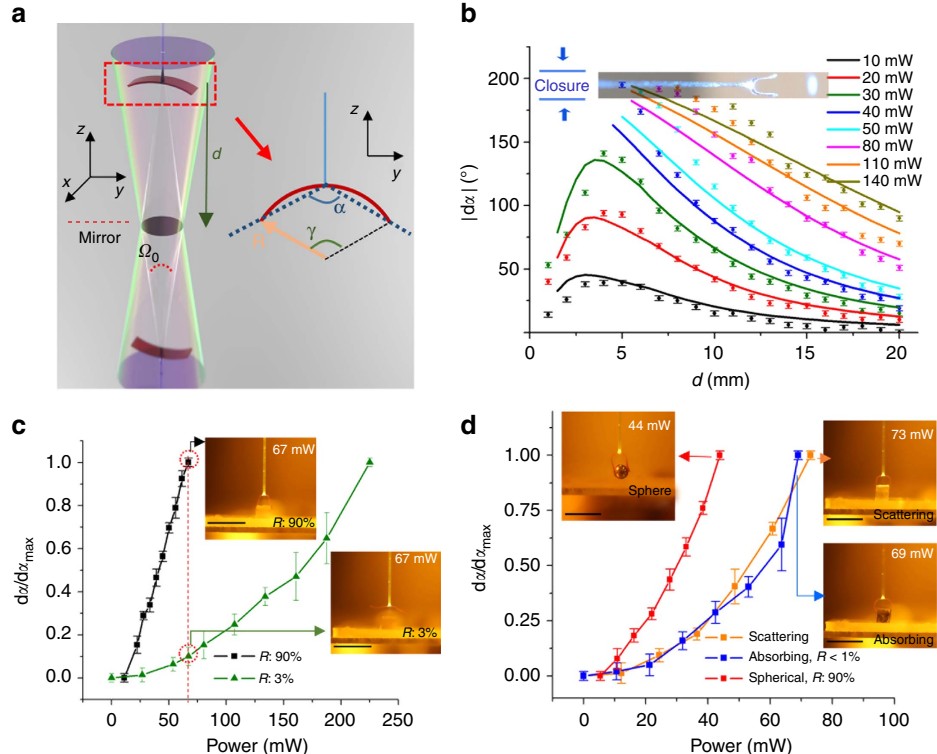

**Figure 3 | Recognition between targets by using feedback-type optical actuation.** (**a**) Schematic drawing of the geometry of the flytrap gripper. (**b**) Change in gripping angle $|d\alpha|$ as a function of distance $d$ at different output powers $P$ (points and lines are experimental and calculated data, respectively). Inset: photograph of the closed gripper with a maximum value in $|d\alpha|$. Error bars indicate the imaging system accuracy (4°) in every single measurement. (**c**) Measured bending ratio $d\alpha/d\alpha_{max}$ as a function of input power $P$ for targets with high ($R = 90\%$) and low ($R = 3\%$) reflectivity. Insets: photographs of the gripper at its closed and open stages when meeting high-reflectivity and low-reflectivity targets, respectively (power 67 mW in both cases). (**d**) Measured bending ratio $d\alpha/d\alpha_{max}$ as a function of input power $P$ for a glass micro-sphere ($R = 90\%$), a highly absorbing ($R < 1\%$) and highly scattering PDMS targets. Insets: photographs of the closed gripper meeting different targets with different threshold powers: 44 mW for the micro-sphere, 73 mW for the scattering and 69 mW for the absorbing target. The error bars in **c** and **d** indicate the imaging system accuracy (4°) plus the s.d. for $n = 3$ measurements. All scale bars correspond to 5 mm. An optical filter is used to block wavelengths below 500 nm for all the photographs.

fibre, and emitted through the center of the device, as shown in Fig. 2f. The transparent window prevents direct light absorption, while all the absorbed light responsible for LCE actuation comes from the reflection/scattering from any encountered object in the probing area. For instance, in front of a highly reflective flat mirror the open gripper (at far distance) performs a gradual closure action when approaching the surface (Fig. 2g). At a fixed distance, the gripping can be manually controlled by changing the output power from the fibre (Fig. 2h, also see Supplementary Movie 1).

**Autonomous action and self-recognition based on optical feedback.** To analyse the performance of the optical flytrap, we measured the gripping angles $\alpha$ (Fig. 3a) as a function of distance $d$ from a mirror at different laser powers $P$. The results are shown in Fig. 3b. Light emits from the fibre tip at a divergence angle $\beta$ [$\beta = 13°$; $NA = n \cdot \sin\beta$, where NA is the numerical aperture of the fibre (0.22) and $n$ is the refractive index of air], which corresponds to a solid angle $\Omega_0$ [$\Omega_0 = 2\pi(1 - \cos\beta) = 0.216$]. Upon reflection from the mirror, the light propagation preserves the same divergence angle, thus the whole illumination space can be considered as a light cone with its apex at the mirror-symmetric position of the fibre tip (Fig. 3a). By approaching the mirror, the distance between the gripper and the light source ($2d$) decreases, that is, the gripper moves towards a region with higher light intensity. Therefore, more light is absorbed by the actuator, leading to smaller $\alpha$ (closing action). For low laser powers ($P < 40$ mW) and at short distances ($d < 5$ mm), the light cone is

too confined to expose the whole actuator, and larger portion of light energy is absorbed in the central area of the LCE where the cured resin prevents actuation of the LCE strip. Thus, the bending deformation saturates and reaches a maximum value at $d \approx 4$ mm. For higher powers, the gripper reaches the closure stage at a certain distance $d_c$, depending on the power used (that is, $d_c = 5$ mm for $P = 40$ mW; $d_c = 12$ mm for $P = 140$ mW, see in Fig. 3b).

The total reflected power equals to $P \cdot R$, where $R$ is the reflectivity of the surface. Thus, the energy distribution of the reflected light takes the form $D(\theta, \varphi)$, indicating the percentage of reflected light in $d\Omega = \sin\theta d\theta d\varphi$ solid angle in spherical coordinates $<r, \theta, \varphi>$. $D(\theta, \varphi)$ strongly depends on the properties of the reflective surface. For a flat mirror, all energy is concentrated within a light cone with solid angle $\Omega = \Omega_0$ (or $D = 0$ for $|\theta| > \beta$). For a concave surface $\Omega < \Omega_0$, for a convex surface $\Omega > \Omega_0$, and for a scattering surface $\Omega >> \Omega_0$. The deformed gripper can be approximated as an arc with a central angle of $\gamma$ (see the inset of Fig. 3a). Thus the change of central angle $d\gamma$ of the gripper can be described as

$$d\gamma = k \cdot E = k \cdot A \cdot P \cdot R \cdot \iint_S D(\theta, \varphi) \frac{\vec{r} d\vec{S}}{r^3} \qquad (1)$$

where $k$ is the light-induced bending coefficient, $A$ is absorption efficiency, $E$ is the total absorbed energy per unit time in the actuated area $S$. The calculated results are shown as lines in Fig. 3b, revealing good consistency between the modelling and the

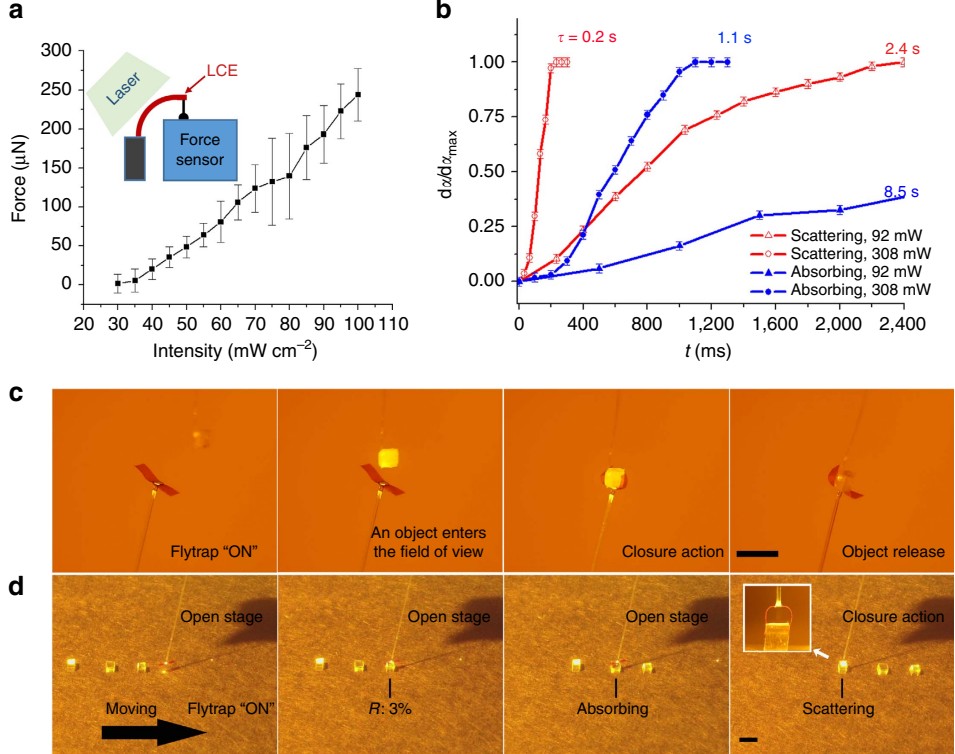

**Figure 4 | Flytrap-type capture motion and applications.** (**a**) Measured force from one LCE actuating arm as a function of illuminated laser intensity. Inset: schematic drawing of the experimental set up. Error bars indicate force sensor accuracy (10 μN) plus the s.d. for $n = 3$ measurements. (**b**) Response time of gripping motion for scattering and absorbing targets by using different laser powers. The error bars indicate the imaging system accuracy (4°) in single measurement. (**c**) The optical flytrap mimics the motion of a natural flytrap by capturing a small scattering object falling on the gripper ($P = 200$ mW). (**d**) Demonstration of self-detection in a moving production line: no response to a transparent cubic (low reflectivity) or an absorbing cubic (long response time), automatic closure when meeting a highly scattering cubic, creating sufficient optical feedback. $P = 150$ mW. Inset: zoom-in side-view optical image of gripping of the scattering cubic. All scale bars are 5 mm. An optical filter is used to block wavelengths below 500 nm for all the photographs.

experimental results. Further details on the modelling can be found in the Supplementary Note 2 and Supplementary Fig. 6.

Equation (1) reveals two working modes for the gripper. Firstly, by changing the light power $P$, the gripper can be manually, yet remotely, controlled. Secondly, for a constant power, the bending angle varies depending on light reflection. The second working mode is useful for developing a smart robotic system, since the power source is free of human control and the device can operate autonomously through adaptation to environmental changes. Hence, we implemented the second mode to demonstrate feedback-type actuation and autonomous action of the device. Figure 3c plots the change of gripping angle d$\alpha$ (normalized as d$\alpha$/d$\alpha_{max}$) as a function of power $P$ for two cubic samples with high and low reflectivity (see Methods for sample fabrication details), demonstrating the distinct actuation behavior in response to different targets. Keeping the same power $P = 67$ mW, the gripper closes when observing the highly reflective micro-cubic, while remaining open when a cubic with low reflectivity is in its field of view (insets of Fig. 3c). The same mechanism holds for other targets with arbitrary reflection/ scattering patterns, resulting in differentiated power thresholds and actuation dynamics for achieving the same closure action. The actuation threshold and dynamics depend on $R$ as well as on physical properties of the object, affecting the specific form of $D(\theta, \varphi)$. As shown in the insets of Fig. 3d, the gripper closes at 44 mW for a silver-coated micro-bead ($d = 2$ mm; $R = 90\%$), and at around 70 mW for a highly scattering cubic and a black, absorbing one ($R < 1\%$), yet with significantly different actuation dynamics. The plots of d$\alpha$/d$\alpha_{max}$ demonstrate a novel

functionality of LCE actuators: at a specific power the device recognizes its favorite prey and acts autonomously to trap it.

**Optical flytrap.** Powered by light, this tiny fibre-tip device can grip on any micro-object with arbitrary shape, from cubic to sphere (Fig. 3c,d), from a scattering particle to a piece of foil (Supplementary Movie 2). The gripping force originates from mechanical bending of the soft LCE actuator, which is proportional to the excitation light intensity (Fig. 4a). The bending force is in the hundreds of μN range and can create sufficient adhesion during the gripping process (statistic friction in the normal direction) that sustains the object weight of tens of mg, which is hundreds of times larger than the mass of the actuator itself. The flytrap gripper can serve as an automatic tool to test different objects, target the desired one, and release it rapidly once the light is turned off. The grip-and-release action is demonstrated in Supplementary Movie 3 for a highly scattering micro-cube with a mass of 10 mg. The response time of the optical flytrap depends on the light power, as shown in Fig. 4b and in Supplementary Fig. 1. By increasing the power to about 300 mW, the device closes within 200 ms. As a comparison, natural flytrap snaps in 100 ms, after about 0.7 s delay between the trigger and the snap[21]. Figure 4b also reveals the slow actuation of the black target, which is dominated by heat transfer. Hence, the scattering and absorbing targets are distinguished by the device through different dynamics, even if the actuation threshold (Fig. 3d) is similar for both. The optical flytrap can capture artificial insects (particles) that enter its field of view. Figure 4c and

Supplementary Movie 4 demonstrate this behavior, showing the optical flytrap capturing a piece of rice falling on the device. The device may also potentially be used in manufacturing units that require automatic detection of product defects. We demonstrate this potential in Fig. 4d, where a set of cubic particles are sitting on a production line (a translation stage) moving below the optical flytrap. Once a product that provides sufficient optical feedback (as demonstrated by using a strongly scattering cubic particle) reaches the flytrap, it is automatically selected (Supplementary Movie 5).

## Discussion

There are two features in the actuation process that we would like to particularly point out. The first one is closure when meeting a strongly absorbing, unreflective target (Fig. 3d). For a black cubic which reflects less than 1% of light, the optical flytrap closes with a threshold much lower than for a non-absorbing one with 3% reflectance. This can be attributed to photothermal heating of the absorbing object, and subsequent heat transfer to the actuator through the air layer, leading to gripping action. This has been confirmed by imaging the thermal distribution of the system with an infrared camera, as shown in the thermal image in Supplementary Fig. 2. Compared with the conventional actuation scheme where the flytrap absorbs the energy directly from the reflected light, the heat-transfer-based actuation takes place at a much longer time scale (Supplementary Movie 6, Fig. 4b). The second phenomenon to point out is the scaling effect in the power threshold. We have recorded the bending for grippers with different sizes (length 4, 6, 8, 10 mm; width 1 mm). As shown in Supplementary Figs 3,4, the thresholds for different objects scale differently with actuator length. For the black target the threshold is significantly reduced for grippers with smaller sizes, due to enhanced heat transfer efficiency at a reduced distance. The size-dependent character of the gripper may provide an additional degree of freedom to design and optimize the performance of the autonomous actuator. We note also that it is relatively straightforward to scale up the size of the device at least to several centimeters simply by using larger liquid-crystal cells. Masked exposure[40] and direct laser writing[41], in turn, should allow for fabricating miniature, microscopic-scale gripping devices. The working distance of the device can be controlled by changing the numerical aperture of the optical fibre used.

The performance of the optical flytrap is reversible, i.e., it can perform grip-and-release cycles repeatedly without decrease in efficiency. However, upon high-power illumination (for example, in Fig. 4c) needed for fast actuation, photothermal heating can soften the material, thereby enhancing adhesion between the actuator and the target. After the illumination is ceased and the temperature decreases, adhesion poses a barrier for material relaxation, leading to irreversibility in the actuation process. However, no permanent damage is caused to the material during this process, and its original shape and performance can be retained by annealing the sample at 45 °C. In some cases, the device performance may also be influenced by non-uniform scattering pattern caused by the target. This non-uniform illumination can generate asymmetric actuation between the two gripper arms. An example of this is shown in Supplementary Movie 7, where a curved surface of a micro-bead provides an asymmetric reflected light field, resulting to different LCE deformation at different positions.

More generally, robotics employing smart materials such as LCEs has seen huge progress during the past decade. However, the research focus has mainly lied in sophisticated control over the deformation by modification of the actuator material[42,43] or in using stimulated sources and complex light patterns[15,29]. We propose two potential trends for future research. Firstly,

feedback-type actuation allows one to use constant illumination to obtain complex deformation determined by the environment. We demonstrate this in our artificial flytrap, where the action is powered by the light reflected/scattered from nearby objects, as opposed to using the light-triggered deformation as a sensor for detecting environmental variation[44]. Secondly, the actuation process can affect the absorption of light that fuels the photomechanical motion, which in return provides feedback and modulates the original actuation[18]. We expect such feedback-type actuators to become pertinent in intelligent micro-robotic systems, therefore bringing novel alternatives for soft-robotic technologies.

In conclusion, we have demonstrated a light-powered gripping device, an artificial flytrap, capable of mimicking the behavior of a natural flytrap. An optical fibre is used to deliver the light energy needed to deform a liquid-crystal elastomer micro-actuator, whose operation is triggered by reflected or scattered light harvested from the environment. Such reflection- or scattering-induced deformation is further applied to induce feedback-type actuation, to obtain autonomous recognition and distinction between different micro-objects. The fibre-tip gripper is a miniature system demonstrating a self-regulating, optically driven device, which may provide a pathway towards autonomous, intelligent micro-robotics.

## Methods

**Materials.** The micro-actuator consists of nematic LCE network polymerized from a mixture containing 77 mol% of LC monomer 4-Methoxybenzoic acid 4-(6-acryloyloxyhexyloxy)phenyl ester (Synthon chemicals), 20 mol% of LC crosslinker 1,4-Bis-[4-(3-acryloyloxypropyloxy)benzoyloxy]-2-methylbenzene (Synthon chemicals), 2 mol% of light-responsive molecule N-Ethyl-N-(2-hydroxyethyl)-4-(4-nitrophenylazo)aniline (Disperse Red 1, Sigma Aldrich), and 1 mol% of photo-initiator (2,2-Dimethoxy-2-phenylacetophenone, Sigma Aldrich). All molecules were used as received. Drops of UV glue (UVS 91, Norland Products INC., Cranbury, NJ) were used as transparent windows in the center of LCE strips and as mechanical connectors between the LCE and the multimode optical fibre (0.22 NA, 200 μm core, Thorlabs).

**Sample preparation.** For LC cell fabrication, two glass slides were firstly spin coated with 1 wt% water solution of polyvinyl alcohol (PVA Sigma Aldrich; 4,000 r.p.m., 1 min) and homeotropic alignment layer (JSR OPTMER, 6,000 r.p.m., 1 min), respectively. The PVA-coated glass slide was rubbed unidirectionally by using a satin cloth, and subsequently blowed with high-pressure nitrogen to remove any dust particles from the surfaces. Tiny drops of UV glue (few nl each) were picked up by a sharp needle and placed onto the PVA-coated slide, after which the slide with the homeotropic alignment layer was glued on top of the PVA-coated one using 20 μm spacers (Thermo scientific) for defining the cell thickness. A UV LED (Thorlabs; 20 mW cm$^{-2}$, 375 nm, 1 min) was used to cure the glue. The monomer mixture was prepared by magnetically stirring the LC mixture at 70 °C (100 r.p.m.) for 1 hour. Then the mixture was infiltrated into the cell on a heating stage at 70 °C and cooled down to 30 °C with a rate of 5 °C min$^{-1}$, to reach splayed nematic alignment. Another UV LED (Prior Scientific; 150 mW cm$^{-2}$, 385 nm, 1 min) was used to polymerize the LC mixture. The cell was opened, and LCE strips were cut out from the sample film by using a blade.

For attaching the optical fibre, firstly the cladding layer was removed, after which the fibre was cut by using a scribe. Then the fibre tip was dipped into a drop of UV glue, lifted up, and placed perpendicularly above the LCE strip. Approaching the fibre tip towards the LCE strip was done using a vertical translation stage until the glue became in contact with the LCE center (connected with the transparent window). A UV LED (375 nm, > 200 mW cm$^{-2}$, 10 s) was used to stabilize the connection.

Elastomeric micro-cubes (PDMS) were fabricated by mixing SYLGARD(R) 184 silicone elastomer base with 10 wt% of its curing agent and solidified at 80 °C for 5 h in a cell geometry with 2-mm-gap. 2 × 2 × 2 mm cubes were cut out from the PDMS after opening the cell. Reflective layer on PDMS cubics and micro-spheres (Fig. 3) was made by deposition of 100 nm silver in a metal evaporator (Edwards Auto 306), yielding reflectivity of ∼90%. Scattering and absorbing layers were obtained by placing 2 × 2 mm-sized pieces of white paper and black foil, respectively, on top of the PDMS cubes.

**Characterization.** Absorption spectra were measured with a UV-Vis spectrophotometer (Cary 60 UV-Vis, Agilent Technologies) equipped with a custom-made polarization controller. Optical images and movies were recorded by using a Canon 5D Mark III camera with a 100 mm lens, and thermal images were recorded

with an Infrared camera (FLIR T420BX) equipped with a close-up $2\times$ lens. Light from a continuous-wave linearly polarized laser (488 nm, Coherent Genesis CX SLM) was coupled into the fibre from one end, and output power from the LCE center at the other end was measured and reported as the excitation power for all the experiments. Light emitted from the fibre is de-polarized with polarization ratio of ∼1.7. No visible actuation difference in LCE has been observed by change of input laser polarization. Before each experiment, the LCE strips were alternately immersed into water and ethanol, to remove surface charge potentially generated upon removing the strips from the glass cells, or upon interacting with objects such as PDMS. The force generation upon photoactuation was measured by mounting a $5\times1\times0.02$ mm$^3$ LCE strip onto a three-dimensional translation stage, and illuminating with 488 nm laser at 45° angle of incidence, as shown in the inset of Fig. 4a. Upon illumination the sample bent over 90° and being in contact with a force sensor, the light-induced force vs. light intensity was recorded.

**Data availability.** The data that support the findings of this study are available from the corresponding author upon request.

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

## Acknowledgements

A.P. gratefully acknowledges the financial support of the European Research Council (Starting Grant project PHOTOTUNE; Agreement No. 679646). O.M.W. is thankful to the graduate school of Tampere University of Technology (TUT), and H.Z. to the TUT postdoctoral fellowship program, for supporting this work. We are indebted to Dr V. Manninen and Dr M. Virkki for assistance with silver deposition and spectral measurements. Dr P. Wasylczyk (Warsaw University), and Prof. Olli Ikkala (Aalto University) are acknowledged for inspiring discussions and insightful comments.

## Author contributions

H.Z. and A.P. conceived the project; O.M.W. and H.Z. performed experiments. All authors contributed in writing the manuscript.

## Additional information

**Competing interests:** The authors declare no competing financial interests.

