## [Peer Review File · Nature Communications]

Reviewers' comments:

Reviewer #1 (Remarks to the Author):

This contribution details the fabrication and performance of an optical device employing the photomechanical response of guest-host polymeric material. The authors have designed the device to emulate the actuation occurring within a Venus flytrap. The work, when published, will likely draw some attention due to the clear bioinspiration.

The English is loose in many places and should be improved. An example - the authors choice of the word "stripe" to describe the geometry of their specimens. I believe they mean "strip". The manuscript is overly declarative with too much hyperbole for my taste – an example of this is the last two sentences of the opening paragraph.

Ignoring the bioinspired device design and performance, the work employs an optical fiber to trigger the response. The authors mention this is the "first time" this has been done. No doubt, this is the first time this design has been realized – but other groups have used optical fibers for many years; specifically Kuzyk et. al. Considering the core novelty of this work is the device design and performance, one recommendation for the authors in review would be to considerably strengthen the illustration of the concept in Figure 1 and the accompanying discussion. I spent considerable time examining this diagram to fully appreciate the design. A better illustration would help a reader quickly understand the approach.

The materials employed in the paper and the data analysis are standard with respect to peer literature. Nothing new, nothing surprising.

With regard to the potential impact of this paper - this paper will likely draw attention to the journal and to this topic. However, while the science is sound there is nothing new or unexpected that will influence the peer research community with regard to the materials employed and their mechanical response. In this way, I believe this work, if it were to be published, will have a greater impact on the popular scientific community than it will on the peer research community examining photomechanical responses in materials.

Reviewer #2 (Remarks to the Author):

Wani et al. describe a simple system consisting of an optical fiber and a light-responsive polymer actuator. Using this system, the authors effectively demonstrate a simple soft robot capable of actuating in response to detected changes in environmental conditions. This change in environmental condition, the proximity of a room temperature object, induces a change in the propagation of light from the optical fiber, which in turn generates shape change in the polymeric material. The bending of the polymer film is then used as a rudimentary gripper, which approximates the behavior of the Venus Flytrap. The work demonstrates the role of feedback in creating next generation robotic components and provides an example of integrating liquid crystal elastomers into functional systems. The reviewer believes that this work will be of impact in the smart materials and robotics communities. However, there are several concerns that should be addressed before publication.

1. The authors state that this work represents "the first attempt to implement feedback-type actuation in soft micro-robotics". Many stimuli-responsive device paradigms have proposed using this generic mechanism, albeit in distinct ways. For example, a recent review presents many of these prior efforts [10.1016/j.copbio.2016.02.032](https://doi.org/10.1016/j.copbio.2016.02.032). At a minimum, the authors should clarify these remarks to better reflect the body of work in stimuli responsive materials.

2. It seems likely that many potential object shapes would generate non-uniform illumination

patterns. Presumably this non-uniform illumination profile would significantly alter the actuation. This potential limitation should be mentioned in the discussion.

3. The authors use the word "snap" to describe the bending of the actuator. This is somewhat misleading as the snapping of the Venus Flytrap refers to the snap-through actuation of the plant as the geometry traverses a mechanical instability. As actuation in this case is limited to bending, this clarification should be made.

4. The authors state "The force is in the hundreds of μN range and can be applied to lift up micro-objects of tens of mg in mass, which is hundreds of times larger than the mass of the actuator itself". While the authors demonstrate the ability to grip objects, this sentence implies that the actuator performs work on the object (lifting). This sentence should be clarified, or data should be presented regarding work performed by the actuator.

5. Some information regarding the sample preparation and illumination conditions is missing, which is necessary to reproduce the work. Was the LCE film attached such that the homeotropic or planar face was glued to the fiber? How is the light emanating from the fiber polarized?

Reviewer #3 (Remarks to the Author):

In the paper, the authors demonstrated an autonomous bending actuation using liquid crystal elastomer (LCE). In the experiment, a LCE film is glued onto the end of an optical fiber. The bending actuation of the LCE film can be triggered by the light reflected from the target object. Consequently, autonomous recognition is realized in the actuating system. I think the work is generally interesting and novel. I would like to recommend the paper to be published in Nature communications. I have the following comments for the authors to further improve the article.

1. I am not very clear about the connection between the plant flytrap and the LCE flytrap. Especially, I do not think the LCE flytrap mimicked the motion of plant flytrap. It has been known that the fast closure of plant flytrap is due to snap-through elastic instabilities, while the LCE flytrap simply bends due to inhomogeneous contraction. These two motion modes, in my opinion, are very different.

2. In the supplemental video 4, one end of LCE gripper seems not reversibly go back to its initial shape after the object is removed. For the applications, the reversibility of actuation can be critical. Can the authors comment on this?

3. Is that possible to scale up or down the LCE flytrap?

4. In Figure 4d, the closure action at the very bottom right is hard to see. It would be good to show the image by zooming in the LCE flytrap.

Response to reviewer comments for manuscript:

“A light-driven artificial flytrap” (NCOMMS-17-02594-T)

Reviewer 1:

Reviewer: This contribution details the fabrication and performance of an optical device employing the photomechanical response of guest-host polymeric material. The authors have designed the device to emulate the actuation occurring within a Venus flytrap. The work, when published, will likely draw some attention due to the clear bioinspiration.

OUR ANSWER: We thank the reviewer for his/her positive comments regarding the novelty and clear bioinspiration, and share his/her view that our work should draw attention from the scientific community and even from the general public.

Reviewer: The English is loose in many places and should be improved. An example - the authors choice of the word “stripe” to describe the geometry of their specimens. I believe they mean “strip”. The manuscript is overly declarative with too much hyperbole for my taste – an example of this is the last two sentences of the opening paragraph.

OUR ANSWER: We agree that we have made mistake in using “stripe” instead of “strip”, and this has been corrected in the revised manuscript. We have also carefully gone through the language throughout the manuscript, examining “the taste” of the sentences, and modifying them when deemed necessary. As examples,

1. The first two sentences of the abstract now read: “*Mimicking natural flytraps, this artificial flytrap is capable of autonomous closure and object recognition. It enables self-regulated actuation within the fiber-sized architecture, thus opening up new avenues towards soft, autonomous small-scale devices.*”
2. The last sentence of the first paragraph of the introduction has been changed and now reads: “*However, incorporation of complex computing circuitry, power sources, and electrically driven actuators into miniaturized robotic systems is challenging, and other approaches are needed to devise smart robotic actuators.*”

Reviewer: Ignoring the bioinspired device design and performance, the work employs an optical fiber to trigger the response. The authors mention this is the “first time” this has been done. No doubt, this is the first time this design has been realized – but other groups have used optical fibers for many years; specifically Kuzyk et. al. Considering the core novelty of this work is the device design and performance, one recommendation for the authors in review would be to considerably strengthen the illustration of the concept in Figure 1 and the accompanying discussion. I spent considerable time examining this diagram to fully appreciate the design. A better illustration would help a reader quickly understand the approach.

Following the reviewer’s advice, we have strengthened the schematic illustration of our flytrap (Figure 1c,d), the caption of Figure 1, as well as the accompanying discussion. In particular, we

have significantly revised the second paragraph in the section “System concept” describing the device, which now reads:

“In order to make this bending deformation to “sense” the environment, we integrated the LCE actuator with an optical fiber. More specifically, we fabricated the splay-aligned LCE actuator onto the tip of the fiber, leaving a transparent window in the center, through which light is emitted. The emission cone determines the “field of view” of the device, which continuously probes the space in front (Fig. 1c). When an object enters into the field of view and produces enough optical feedback (reflected/scattered light), the LCE bends towards the object (closure action), eventually capturing it (Fig. 1d). The optical feedback is determined by the reflectance/scattering intensity of the object, and therefore, the artificial flytrap may exhibit distinct actuation behavior when meeting different targets. Compared to other flytrap-like devices reported to date^{36,37}, this is the first miniature device that mimics the intelligent features of the Venus flytrap, while the mechanical motion is triggered by light-induced bending of the LCE actuator, not elastic instability²¹ as in the case of the Venus flytrap.”

We believe that in the revised form, the device design and operation should be clear.

We would also like to mention that we are well aware on large scientific and technological activities around optical fibers, including the significant contributions of the group of Mark Kuzyk. The Kuzyk group has also devised light-responsive, dye-doped optical fibers that can activate a cantilever under laser radiation [J. Opt. Soc. Am. B 2006, 23, 697]. The goal of that work, as well as the materials design, is completely different from our approach. To acknowledge the significant contribution of the Kuzyk group to photomechanical materials, we added two of his references ([31,32]) that we deem most relevant for the present work, into the reference list. We also modified the last sentence of the “System concept”, which now reads:

“Compared to other flytrap-like devices reported to date^{36,37}, this is the first miniature device that mimics the intelligent features of the Venus flytrap, while the mechanical motion is triggered by light-induced bending of the LCE actuator, not elastic instability²¹ as in the case of the Venus flytrap.”

***Reviewer:** The materials employed in the paper and the data analysis are standard with respect to peer literature. Nothing new, nothing surprising.*

OUR ANSWER: We acknowledge that the material used in the system is not new, but would like to emphasise that our focus lies in novel (bioinspired) device design, not in novel materials. Instead, we found it more useful to use materials that are easily accessible through commercial sources. As we use standard actuator materials, we also use standard analysis (bending angle, response speed, force, etc.) to characterize their performance and find good agreement between experiments and modelling as presented in Figure 3, confirming that our device operates as planned.

***Reviewer:** With regard to the potential impact of this paper - this paper will likely draw attention to the journal and to this topic. However, while the science is sound there is nothing new or unexpected that will influence the peer research community with regard to the materials employed and their mechanical response. In this way, I believe this work, if it were to be published, will have*

a greater impact on the popular scientific community than it will on the peer research community examining photomechanical responses in materials.

OUR ANSWER: As detailed in the previous answer, there is indeed nothing new in the material used, which is deliberate since the point of the article lies elsewhere. Our biggest achievement is the biomimetic device design, providing autonomous, feedback-based operation, and ability to distinguish between objects. We believe that the work will be of interest across scientific borders and to general public.

We disagree with the reviewer regarding his/her comment that our result will not influence the research community working with photomechanics. This is because no research has been done on LCE robotic systems that autonomously react to environmental changes (in our case, change in light reflection/scattering) or use feedback mechanism to power robotic actuation. We believe our artificial optical flytrap is the first demonstration on this concept. Please note that such feedback-type actuation is previously unexplored in the context of LCE photomechanics. Herein, the feedback is provided by light reflection/scattering, but also many other types of feedback mechanisms, based *e.g.* on light polarization or emission, can be envisaged. Because of this, we believe our work to open up multitude of possibilities and to attract the attention of the LCE community and beyond.

Reviewer #2:

***Reviewer:** Wani et al. describe a simple system consisting of an optical fiber and a light-responsive polymer actuator. Using this system, the authors effectively demonstrate a simple soft robot capable of actuating in response to detected changes in environmental conditions. This change in environmental condition, the proximity of a room temperature object, induces a change in the propagation of light from the optical fiber, which in turn generates shape change in the polymeric material. The bending of the polymer film is then used as a rudimentary gripper, which approximates the behavior of the Venus Flytrap. The work demonstrates the role of feedback in creating next generation robotic components and provides an example of integrating liquid crystal elastomers into functional systems. The reviewer believes that this work will be of impact in the smart materials and robotics communities. However, there are several concerns that should be addressed before publication.*

OUR ANSWER: We are thankful to the reviewer for his/her comments about the novelty of our research, and its potential impact in smart materials and robotics communities.

***Reviewer:** The authors state that this work represents “the first attempt to implement feedback-type actuation in soft micro-robotics”. Many stimuli-responsive device paradigms have proposed using this generic mechanism, albeit in distinct ways. For example, a recent review presents many of these prior efforts [10.1016/j.copbio.2016.02.032](https://doi.org/10.1016/j.copbio.2016.02.032). At a minimum, the authors should clarify these remarks to better reflect the body of work in stimuli responsive materials.*

OUR ANSWER: The term “robot” used in this manuscript refers to the conventional robot concept – a device that can move and has robotic function like locomotion, gripping, and so on. The materials employed to devise soft robots comprise, e.g., artificially patterned biological tissue, pneumatic actuators, LC elastomers, and hydrogels. References 12-19 in the manuscript provide examples of these different types of stimuli-responsive materials used in soft robotics. As for “being the first”, to the best of our knowledge we demonstrate the first soft-robot actuation device operated by feedback from the environment.

We do not consider the dynamic biomaterials (nanoscopic or microscopic particles), pointed out by the reviewer, to fall into the category of soft micro-robotics. At the same time, we acknowledge that it is important to bring out that autonomous feedback has been applied in such systems, and therefore add the reference proposed by the reviewer to the citation list of the revised manuscript (Ref. [3]).

***Reviewer:** It seems likely that many potential object shapes would generate non-uniform illumination patterns. Presumably this non-uniform illumination profile would significantly alter the actuation. This potential limitation should be mentioned in the discussion.*

OUR ANSWER: We thank the reviewer for this very relevant comment. To address the point raised, we have added a Supplementary Movie and the following text into “Discussion”:

“In some cases, the device performance may also be influenced by non-uniform scattering pattern caused by the target. This non-uniform illumination can generate asymmetric actuation between the two gripper arms. An example of this is shown in Supplementary Movie 7, where a

curved surface of a micro-bead provides an asymmetric reflected light field, resulting to different LCE deformation at different positions.”

Reviewer: *The authors use the word “snap” to describe the bending of the actuator. This is somewhat misleading as the snapping of the Venus Flytrap refers to the snap-through actuation of the plant as the geometry traverses a mechanical instability. As actuation in this case is limited to bending, this clarification should be made.*

OUR ANSWER: In the process of writing the manuscript, we were thinking a lot whether it is justified to use the word “snap” in connection to our LCE actuator, and were finally inclined to use it even if our actuator is not driven by mechanical instabilities alike the Venus flytrap. Since both Reviewer 2 and Reviewer 3 point out that this is misleading, we realize that this was a misjudgement. Therefore, we have removed the word “snap” to describe the actuation of our artificial flytrap, and replaced it with “closure”, to avoid confusions. We have also added the following clarifying sentence to the end of “System concept”:

“Compared to other flytrap-like devices reported to date^{36,37}, this is the first miniature device that mimics the intelligent features of the Venus flytrap, while the mechanical motion is triggered by light-induced bending of the LCE actuator, not elastic instability²¹ as in the case of the Venus flytrap.”

Reviewer: *The authors state “The force is in the hundreds of μN range and can be applied to lift up micro-objects of tens of mg in mass, which is hundreds of times larger than the mass of the actuator itself”. While the authors demonstrate the ability to grip objects, this sentence implies that the actuator performs work on the object (lifting). This sentence should be clarified, or data should be presented regarding work performed by the actuator.*

OUR ANSWER: We agree that this sentence in its original form is misleading, and we have adjusted it to read as follows:

“The bending force is in the hundreds of μN range and can create sufficient adhesion during the gripping process (static friction in the normal direction) that sustains the object weight of tens of mg, which is hundreds of times larger than the mass of the actuator itself.”

Reviewer: *Some information regarding the sample preparation and illumination conditions is missing, which is necessary to reproduce the work. Was the LCE film attached such that the homeotropic or planar face was glued to the fiber? How is the light emanating from the fiber polarized?*

OUR ANSWER: As shown in Figure 1, the actuator is glued onto the fiber tip such that homeotropic face is on the fiber side. If the sample was glued the opposite way, no gripping action could be achieved. We added one sentence in the “System realization” to make this clear. As for light polarization, the input laser is linearly polarized, while the light emitted from the multi-mode optical fiber (diameter 200 μm , length > 2m) is de-polarized with polarization ratio of ~ 1.7 . Due to the high absorbance of the LCE film, no visible actuation differences could be detected by change of input laser polarization, as is now clarified in the Methods section (p. 13).

Reviewer #3:

Reviewer: In the paper, the authors demonstrated an autonomous bending actuation using liquid crystal elastomer (LCE). In the experiment, a LCE film is glued onto the end of an optical fiber. The bending actuation of the LCE film can be triggered by the light reflected from the target object. Consequently, autonomous recognition is realized in the actuating system. I think the work is generally interesting and novel. I would like to recommend the paper to be published in Nature communications. I have the following comments for the authors to further improve the article.

OUR ANSWER: We thank the reviewer for his/her positive assessment.

Reviewer: I am not very clear about the connection between the plant flytrap and the LCE flytrap. Especially, I do not think the LCE flytrap mimicked the motion of plant flytrap. It has been known that the fast closure of plant flytrap is due to snap-through elastic instabilities, while the LCE flytrap simply bends due to inhomogeneous contraction. These two motion modes, in my opinion, are very different.

OUR ANSWER: We agree with the reviewer that it is misleading to use the word “snap” in the context of our LCE actuator. Therefore, we have removed this word, using the word “closure” instead, and have included the below sentence the end of “System concept” in order to clarify that the actuation mechanism of our flytrap is different from the venus flytrap:

“Compared to other flytrap-like devices reported to date^{36,37}, this is the first miniature device that mimics the intelligent features of the Venus flytrap, while the mechanical motion is triggered by light-induced bending of the LCE actuator, not elastic instability²¹ as in the case of the Venus flytrap.”

We point out that our device combines three important features inherent to the Venus flytrap: (i) automatic closure upon external stimulus, (ii) sub-second scale fast actuation, and (iii) ability to distinguish between objects. Therefore, we believe it is justified to use the word “artificial flytrap” to describe its performance, even if the action is not driven by snapping motion.

Reviewer: In the supplemental video 4, one end of LCE gripper seems not reversibly go back to its initial shape after the object is removed. For the applications, the reversibility of actuation can be critical. Can the authors comment on this?

OUR ANSWER: We thank the reviewer for bringing out this important point. In general, the actuation of the gripper is fully reversible, and we can grip and release objects several times. This is evident from Supplementary Video 3, where a scattering object is gripped and released, and symmetric unbending during release is clearly visible. In the Supplementary Video 4, our aim was to demonstrate that we can also achieve very fast actuation if we use sufficiently high powers. And at high powers, photothermal heating softens the material and enhances adhesion between the actuator and the object. After light is turned off and the material cools down, the adhesion poses a barrier for material relaxation, which is the cause for the observed asymmetry. However, the actuator itself is not damaged, and the original symmetric shape (and original actuation behavior) can be retained by heating the actuator (to 45 °C), or by light irradiation. This is now made clear in the text (Discussion, p. 10), reading as follows:

“The performance of the optical flytrap is reversible, i.e., it can perform grip-and-release cycles repeatedly without decrease in efficiency. However, upon high-power illumination (e.g. in Fig. 4c) needed for fast actuation, photothermal heating can soften the material, thereby enhancing adhesion between the actuator and the target. After the illumination is ceased and the temperature decreases, adhesion poses a barrier for material relaxation, leading to irreversibility in the actuation process. However, no permanent damage is caused to the material during this process, and its original shape and performance can be retained by annealing the sample at 45 °C.”

Reviewer: 3. *Is that possible to scale up or down the LCE flytrap?*

OUR ANSWER: The device can be scaled up relatively easily, even if polymer networks with larger elastic modulus might be needed to sustain the flat shape of the device in its non-actuated form. By changing the numerical aperture of the optical fiber, the working distance can be controlled. The size of the device can also be scaled down if desired, for instance by using direct laser writing techniques to fabricate the actuator onto the fiber tip. To address the scaling issues, we have added the following text to the Discussion section (p. 10):

“We note also that it is relatively straightforward to scale up the size of the device at least to several centimeters simply by using larger liquid-crystal cells. Masked exposure⁴⁰ and direct laser writing,⁴¹ in turn, should allow for fabricating miniature, microscopic-scale gripping devices. The working distance of the device can be controlled by changing the numerical aperture of the optical fiber used.”

Reviewer: 4. *In Figure 4d, the closure action at the very bottom right is hard to see. It would be good to show the image by zooming in the LCE flytrap.*

OUR ANSWER: Following the reviewer’s advice, we have added a zoomed-in image of the actuator in the closed stage into Figure 4d, based on which the closure action on the scattering cubic is now very clear.

REVIEWERS' COMMENTS:

Reviewer #1 (Remarks to the Author):

In reading through the response to my previous comments as well as those of the other reviewers, I believe the authors have appropriately addressed my prior concerns.

Reviewer #2 (Remarks to the Author):

The authors have sufficiently addressed concerns presented by each of the reviewers. In light of these revisions, it is my recommendation that the work be published in its current form.

Reviewer #3 (Remarks to the Author):

I am happy about the current version of the paper. Therefore, I recommend the paper to be published in nature communication.